# Disentangling ERBB Signaling in Breast Cancer Subtypes—A Model-Based Analysis

**DOI:** 10.3390/cancers14102379

**Published:** 2022-05-12

**Authors:** Svenja Kemmer, Mireia Berdiel-Acer, Eileen Reinz, Johanna Sonntag, Nooraldeen Tarade, Stephan Bernhardt, Mirjam Fehling-Kaschek, Max Hasmann, Ulrike Korf, Stefan Wiemann, Jens Timmer

**Affiliations:** 1Institute of Physics, University of Freiburg, 79104 Freiburg, Germany; svenja.kemmer@fdm.uni-freiburg.de (S.K.); mirjam.fehling-kaschek@posteo.de (M.F.-K.); 2FDM—Freiburg Center for Data Analysis and Modeling, University of Freiburg, 79104 Freiburg, Germany; 3Division of Molecular Genome Analysis, German Cancer Research Center, 69120 Heidelberg, Germany; mireia.berdiel@gmail.com (M.B.-A.); eileen.reinz@gmail.com (E.R.); johanna.nelly.sonntag@gmail.com (J.S.); n.tarade@dkfz-heidelberg.de (N.T.); stephanbernhardtemail@gmail.com (S.B.); d.fischer@dkfz.de (U.K.); 4Faculty of Biosciences, University of Heidelberg, 69117 Heidelberg, Germany; 5Roche Diagnostics, 82377 Penzberg, Germany; max.hasmann@web.de; 6Signalling Research Centres BIOSS and CIBSS, University of Freiburg, 79104 Freiburg, Germany

**Keywords:** ERBB signaling, targeted therapy, systems biology, mathematical modeling, breast cancer, signal transduction

## Abstract

**Simple Summary:**

Breast cancer subtypes are characterized by the expression and activity of estrogen-, progesterone- and HER2-receptors and differ by the treatment as well as patient prognosis. Tumors of the HER2-subtype overexpress this receptor and are successfully targeted with anti-HER2 therapies. We wanted to know if the HER2-receptor and the downstream signaling network act similarly also in the other subtypes and if this network could potentially be a therapeutic target beyond the HER2-positive subtype. To this end, we quantitatively assessed the wiring of signaling events in the individual subtypes to unravel the characteristics of HER-signaling. Our data along with a model-based analysis suggest that major parts of the intracellular signal transduction network are unchanged between the different breast cancer subtypes and that the clinical differences mostly come from the different levels at which these receptors are present in tumor cells as well as from the particular mutations that are present in individual tumors.

**Abstract:**

Targeted therapies have shown striking success in the treatment of cancer over the last years. However, their specific effects on an individual tumor appear to be varying and difficult to predict. Using an integrative modeling approach that combines mechanistic and regression modeling, we gained insights into the response mechanisms of breast cancer cells due to different ligand–drug combinations. The multi-pathway model, capturing ERBB receptor signaling as well as downstream MAPK and PI3K pathways was calibrated on time-resolved data of the luminal breast cancer cell lines MCF7 and T47D across an array of four ligands and five drugs. The same model was then successfully applied to triple negative and HER2-positive breast cancer cell lines, requiring adjustments mostly for the respective receptor compositions within these cell lines. The additional relevance of cell-line-specific mutations in the MAPK and PI3K pathway components was identified via L_1_ regularization, where the impact of these mutations on pathway activation was uncovered. Finally, we predicted and experimentally validated the proliferation response of cells to drug co-treatments. We developed a unified mathematical model that can describe the ERBB receptor and downstream signaling in response to therapeutic drugs targeting this clinically relevant signaling network in cell line that represent three major subtypes of breast cancer. Our data and model suggest that alterations in this network could render anti-HER therapies relevant beyond the HER2-positive subtype.

## 1. Introduction

The development of therapeutic antibodies and small-molecule kinase inhibitors has revolutionized cancer treatment. Many of these therapeutics target receptor tyrosine kinases, which connect extracellular growth factors with intracellular signaling networks and, when deregulated, drive oncogenic mechanisms [1]. Advances in understanding the complex signaling network regulating tumor progression have facilitated the development and effective use of targeted therapies [2]. Although this field emerged in 1998 with the FDA approval of the first monoclonal antibody for breast cancer treatment, trastuzumab [3], the overall therapy efficacy has remained unsatisfactory [4]. This appears to be due to the molecular heterogeneity between different breast cancer subtypes as well as between individual tumors [5].

The ERBB signaling network is one of the main drivers of cell growth [2]. The ERBB receptor family, comprising four paralogous receptor tyrosine kinases, EGFR (ERBB1/HER1), ERBB2 (HER2), ERBB3 (HER3) and ERBB4 (HER4), is activated by a variety of growth factors, including the epidermal growth factor (EGF), betacellulin (BTC), transforming growth factor alpha (TGFa) and neuregulin 1 (NRG1) [6,7,8,9]. ERBB receptors share a common structure that is characterized by extracellular ligand binding and intracellular kinase domains, which are separated by a single-pass transmembrane domain.

ERBB2 and ERBB3 are exceptional as the structure of the ERBB2 ligand-binding domain is constitutively in a dimerization-prone conformation rendering the binding of a ligand unnecessary to induce dimerization, while ERBB3 is kinase-impaired [10]. Upon ligand binding, the receptors undergo dynamic conformational changes and homo- or hetero-dimerize. ERBB2 has a preference for heterodimerization with ERBB3; however, the dynamics of dimerization are vastly dependent on the relative abundance of the individual receptors as well as on the presence of their respective ligands. Dimerization triggers auto- and transphosphorylation of the receptors as well as the subsequent activation of downstream signaling pathways.

Among them, the MAP kinase (MAPK) cascade as well as the phosphoinositide 3-kinase (PI3K) pathway play essential roles in signal processing and propagation [11,12,13,14,15]. Given the key function of ERBB receptors in the process of growth regulation, several inhibitory drugs have been developed to target this receptor family. Examples include the monoclonal antibodies cetuximab, trastuzumab, pertuzumab and lumretuzumab as well as the kinase inhibitor erlotinib [16,17,18,19,20]. Cetuximab and erlotinib are both directed against EGFR, trastuzumab and pertuzumab target ERBB2, whereas lumretuzumab inhibits signaling via ERBB3 [21].

Regulation of the ERBB signaling network is highly complex, varies between individual breast cancer subtypes and has been previously studied using mathematical modeling—a powerful tool to render complex systems intelligible and accessible [22,23,24,25,26,27]. Particularly the downstream MAPK signaling cascade is a crucial module within this network and has been a core element of several models [28,29,30,31]. These studies focused on the response of cells to stimulation with different ligands, others also covered the ligand-dependent response dynamics of the ERBB signaling network [27,32,33,34,35].

Furthermore, the drug-response mechanisms of different breast cancer subtypes have been evaluated before. For example, Schoeberl et al. explored the optimal way to therapeutically inhibit ligand-induced activation of the ERBB–PI3K axis [36], while Imoto et al. analyzed the drug sensitivity in the ERBB–MAPK–PI3K signaling network in breast cancer patients and included a proliferation/survival analysis [37].

Here, we investigated combinations of several ligands as well as of different therapeutic drugs and their dynamic effects on cell line models of three breast cancer subtypes and their proliferative behavior. Beyond the previously published work, we expanded the analysis by weighting cell-type-specific differences in model parameters and thus the characteristics of the cell lines that propagate in differential activation of the individual signaling branches.

To this end, we used data-based mechanistic modeling with ordinary differential equations (ODEs) to understand the response mechanisms to different ligands and drugs in three breast cancer subtypes. A signaling model, including receptors as well as downstream signal transduction was developed and calibrated on time-resolved proteomics data. To link the signaling output to the condition-specific growth response of the cells, we implemented a proliferation model applying a linear regression approach.

A proliferation screen of single treatments provided calibration data for the regression parameters, while we could successfully predict proliferation measurements of various drug co-treatments thereby validating our model. Along these lines, we identified, by L_1_ regularization, only two biological specifics that needed to be adjusted to make our model applicable to cell lines of other breast cancer subtypes.

On the one hand, the receptor expression levels were different in the respective cell lines, and on the other hand, the mutation states in genes that are components of the investigated pathways strongly impacted the wiring of signaling. Still, the core signaling network was not affected by these adjustments thus underlining its general applicability. Our study therefore supports the concept of targeting the ERBB signaling network beyond the HER2-positive subtype of breast cancer.

## 2. Materials and Methods

### 2.1. Cell Culture

The human breast cancer cell lines MCF-7 (HTB-22™), T47D (HTB-133™), MDA-MB-231 (HTB-26™) and SKBR3 (HTB-30™) were obtained from the American Type Culture Collection (ATCC, LGC Standards GmbH, Wesel, Germany) and maintained in Dulbecco’s Modified Eagle’s Medium: Nutrient Mixture F-12 (DMEM/F12 without phenol red, Thermo Fisher Scientific, Rockford, IL, USA), supplemented with 10% FCS (Thermo Fisher Scientific, Rockford, IL, USA), 50 units/mL penicillin and 50 µg/mL streptomycin sulfate (Invitrogen AG, Carlsbad, CA, USA) at 37 °C with 5% CO_2_. All cell lines were authenticated (Multiplexion, Heidelberg, Germany) and negatively tested for mycoplasma contamination before and after completion of the study.

### 2.2. Antibodies, Drugs and Ligands

Details on all the antibodies, drugs and ligands used in this study are provided in Appendix A. Therapeutic antibodies were applied at a concentration of 10 μg/mL and the tyrosine kinase inhibitor erlotinib at 1 μM. Ligands were used at a 5 nM concentration.

### 2.3. Time-Resolved Experiments

For time course experiments, 3 × 10${}^{5}$ cells (4 × 10${}^{5}$ for SKBR3) were seeded in each well of six-well plates and starved overnight in DMEM/F12 (Thermo Fisher Scientific, Rockford, IL, USA) without FCS. The following day, cells were treated with drugs for 1 h prior to the addition of the respective ligands. Lysates were collected at the indicated time points using mammalian protein extraction reagent (M-PER™, 78051, Thermo Fisher Scientific, Rockford, IL, USA) except for SKBR3, which was lysed using RIPA buffer (Thermo Scientific™ RIPA Lysis and Extraction Buffer, 89900, Thermo Fisher Scientific, Rockford, IL, USA), all containing protease inhibitor Complete Mini and phosphatase inhibitor PhosSTOP (both Roche Diagnostics, Mannheim, Germany). The protein concentrations were determined by BCA Protein Assay Reagent Kit (Thermo Fisher Scientific, Rockland, IL, USA) and further used for the reverse phase protein array.

### 2.4. Reverse Phase Protein Array (RPPA)

The RPPA experiments were performed as previously described [38,39]. Briefly, cell lysates from three biological replicates for every condition were spotted on nitrocellulose-coated glass slides (Oncyte Avid, Grace-Biolabs, Bend, OR, USA) in technical triplicates each. All antibodies used were previously tested with Western blotting to validate their specificity. The signal intensities of the spots were quantified using GenePixPro 5.0 (Molecular Devices, Sunnyvale, CA, USA). Preprocessing and scaling of the RPPA data, background correction, and the merging of technical triplicates was performed in R using the software packages RPPanalyzer [40] and BlotIt [41].

### 2.5. Viability Assays

To determine the effects of the different ligands and drug treatments on the viability of cancer cells, 3000 cells of the respective cell lines were seeded in 96-well white plates in DMEM/F12 with 10% FCS (all Thermo Fisher Scientific, Rockford, IL, USA). Prior to the addition of ligands, the cells were pre-treated with the corresponding drugs for 1 h. Cells were grown for 6 days, and the effects on cell viability, i.e., the amount of ATP reflecting the number of metabolically active and proliferative cells, was evaluated using CellTiter-Glo^®^ luminescent assay (G7570, Promega, Mannheim, Germany). Luminiscence was determined using a GloMax^®^ microplate reader (GM3000, Promega, Mannheim, Germany).

### 2.6. Mechanistic Modeling

The developed model consisted of two parts, a mechanistic ODE model describing the ERBB signaling network and a linear regression model linking cell proliferation to signaling features. The mechanistic signaling model was described by nonlinear ODEs $\dot{x}\left(t\right)=f\left(x\left(t\right),u\left(t\right),p_{dyn}\right)$ where *x* denoted the vector of dynamic states, such as c-RAF, MEK, etc., and *u* represented external perturbations, such as treatment with a ligand or a drug. Initial concentrations and kinetic rates were comprised in $p_{dyn}$. Model states were linked to experimental measurements by an observation function $y\left(t\right)=g(x\left(t\right),p_{obs})+\epsilon \left(t\right)$ with offset and scaling parameters included in $p_{obs}$ and the assumption of Gaussian errors ϵ∼N(0,σ) achieved by log transformation. Details about model reactions and corresponding observation functions are described in Appendix A.

### 2.7. Parameter Estimation and Uncertainty Analysis

The mechanistic model along with corresponding observations was parameterized by the parameters *p*, including the dynamic model parameters $p_{dyn}$ as well as the observation parameters $p_{obs}$. One model structure was used to describe different experimental conditions, such as the distinct combinations of ligands and drugs. Those parameters that were affected by the ligand or drug were implemented to be condition-specific and were thus estimated independently for every condition. As cells were kept under constant environmental conditions before the measurements, the system could be assumed to be in equilibrium, i.e., in a steady state at t=0. All initial values $x\left(0\right)=x_{0}$ were therefore described by a steady state transformation [42]. The parameters were estimated from time-resolved measurements based on the maximum likelihood method. Therefore, the objective function
(1)$$-2\log\left(L\right)=\chi^{2}\left(p\right)=\sum_{i}^{N_{t}}\left[{\left(\frac{y_{i}^{D}-y_{i}\left(p\right)}{\sigma_{i}}\right)}^{2}+2\log\left(\sigma_{i}\right)\right]$$
with measured data $y_{i}^{D}$, model trajectories $y_{i}\left(p\right)$ and absolute errors $\sigma_{i}$ was minimized to determine the optimal parameter set $\hat{p}$ given in Appendix A. The resulting maximum likelihood estimate was further analyzed to evaluate identifiability and uncertainty of the optimal parameters. We therefore used the profile-likelihood method [43] obtaining confidence intervals for each parameter that were subsequently used for model reduction as described by Maiwald et al. [44]. The model was reduced until all parameters were identifiable (Appendix A).

### 2.8. Regression Modeling

The mechanistic signaling model was linked to cell proliferation by a regression model. Signaling features, including the area under the curve (AUC), the half-life and the peak height, were derived from the calibrated dynamic model for ppERK, pAKT, pP70S6K and active receptor dimers and used as predictor variables for the linear regression model with the cell proliferation as the response variable. Signaling features with the highest predictive power were selected applying an L_1_ penalty [45], and the correlation coefficients are displayed in Appendix A. The final reduced regression model was specified as
(2)$$y=\alpha_{1}+\beta_{1}\cdot {\mathrm{AUC}}_{pAKT}+\beta_{2}\cdot {\mathrm{AUC}}_{pP70S6K}$$
with the proliferation response y, described by the y-intercept $\alpha_{1}$ and the weighted selected predictor variables ${\mathrm{AUC}}_{pAKT}$ and ${\mathrm{AUC}}_{pP70S6K}$. The model was refitted to the experimental single treatment data from the proliferation assay to obtain the final parameter estimates used for the predictions of co-treatments (Appendix A).

### 2.9. Growth Predictions

As previous studies reported synergistic effects of drug co-treatments [46,47,48,49,50], growth predictions for drug co-treatments were generated by replacing the impact of single drugs $\left(\frac{1}{1+k_{drug1}}\right)$ on the formation of active dimers by their respective combinatorial effect $\left(\frac{1}{1+k_{drug1}}\right)\cdot \left(\frac{1}{1+k_{drug2}}\right)$. All parameters, including the drug impact variables, were fixed to the values estimated based on single treatments.

### 2.10. L_1_ Regularization for ODE Models

The identification of cell-line-specific parameters for the cell lines MDA-MB-231 and SKBR3 were performed using a lasso-based regularization approach [51,52,53]. The parameters for the respective additional cell line $p_{i}^{*}$ were related to the parameters determined for T47D and MCF7 $p_{i}$ by $p_{i}^{*}=p_{i}\cdot r_{i}$ with $r_{i}$ indicating the fold changes between the respective cell lines. All fold changes were regularized with the regularization weight λ favoring $log(r_{i})=0$, where the parameter $p_{i}$ was the same in both breast cancer subtypes. For every λ, a multi-start optimization was performed by first applying the L_1_ prior, and subsequently refitting the model without those fold change parameters was estimated to be compatible with 0. The λ selected to determine the final parameter vector was determined based on AIC statistics (Appendix A).

### 2.11. Flux Analysis

While kinetic rates constants $k_{i}$ describe the speed of a reaction, the conversion of reaction educts can be assessed by the reaction flux $v=k_{i}\cdot c_{Educt}$ with the educt concentration $c_{Educt}$. Note that cell systems with the same kinetic rate parameters can still differ in their respective reaction fluxes due to differing protein concentrations in these systems. Here, we analyzed the two reaction fluxes for P70S6K phosphorylation in the different cell lines, activated either via AKT “phospho_S6K_AKT·S6K·pAKT” or via ERK “phospho_S6K_ERK·S6K·ppERK”. By evaluating the log10 ratio of these two fluxes, we quantified which pathway branch dominated the P70S6K activation in the individual cell lines.

## 3. Results

### 3.1. ERBB Downstream Activation Correlates with Receptor Composition

ERBB downstream activation correlates with the receptor composition. To investigate the impact of various growth factors on the dynamic signaling response of cells derived from different breast cancer subtypes, we studied two cell lines of the luminal A subtype, T47D and MCF7, in addition to the triple-negative MDA-MB-231 and HER2-positive SKBR3 cell lines. The cells were stimulated with four ligands to induce receptor activation and downstream signaling in the absence or presence of four drugs targeting EGFR, ERBB2, or ERBB3. EGF, BTC and TGFa are ligands for EGFR. BTC and NRG1, a ligand for ERBB3, also bind and activate ERBB4, another receptor of the ERBB family (Figure 1a). We assumed that ERBB4 would likely not impact our cell line systems due to the low protein levels in all cell lines in our study (Figure 1b).

To experimentally validate this, we inhibited ERBB3 or knocked down ERBB4 and observed that ERBB3 was required for the activation of downstream signaling by NRG1, while ERBB4 was not (Appendix A). We concluded that ERBB4 should not be relevant in our experimental systems and thus did not regard this receptor in the following.

We then quantified the dynamic changes in the phosphorylation states and the total protein levels of eight central proteins in the ERBB–MAPK–PI3K signaling network. In total, we experimentally tested 80 conditions, covering four cell lines, which were incubated in an array with four ligands, five drugs and an untreated control (Figure 1c). These conditions were evaluated at ten time points for three cell lines and at five for SKBR3. Quantification of the targeted proteins and their phospho-sites in biological triplicates for all conditions generated a data set comprising 33,120 data points.

In an initial analysis, we wanted to discern cell-line-specific effects focusing on the response to the ligands EGF and NRG1 by assessing their effect on the activation of AKT and ERK proteins as representatives of the PI3K and MAPK pathways, respectively. These two pathways are the two main downstream signaling branches activated upon EGF/NRG1 stimulation. As shown in Figure 1d, the growth factors induced diverging responses in the different cell lines, which were associated with distinct states of signal intensities and duration.

This indicated cell-line-specific differences in the propagation of the original input signals. To explain the observed differential activation behavior, we first regarded the receptor composition of the four cell lines as described by Niepel et al. [54] (Figure 1b). The receptor repertoire of the luminal cell lines is mainly constituted by ERBB2 and ERBB3, while they still express a considerable amount of EGFR. The triple negative breast cancer (TNBC) cell line MDA-MB-231 is strongly dominated by EGFR. In contrast, the HER2-positive SKBR3 cell line expresses mostly ERBB2.

Considering the ligand specificity of EGF to EGFR and of NRG1 to ERBB3 (and ERBB4), the differential downstream response might at least partially originate from the distinct ratios of the ERBB-receptors expressed by the cell lines. To test this hypothesis, an activation ratio of NRG1 over EGF was calculated for each protein and cell line, based on the area under the curve (AUC) of the phosphorylation dynamics displayed in Figure 1d. Figure 1e illustrates the correlation between the NRG1/EGF activation ratio and the receptor ratio of ERBB3 to EGFR.

Furthermore, there exists a strong dependence of pathway activation on the respective repertoire of receptors. This correlation followed our expectations, as cells with high ERBB3/EGFR ratios, such as MCF7, were more susceptible for NRG1, and conversely MDA-MB-231 was for EGF. Similarly, the horizontal location of pAKT and pERK in the correlation plot reflects the subsequent downstream activation, where AKT is shifted to the right, meaning to an activation via NRG1.

The exception is represented by MDA-MB-231, which predominantly expresses EGFR and, in addition, carries an activating mutation in KRAS all favoring signaling via the MAPK pathway. Hence, our findings confirm the known dominant role of ERBB3 in AKT activation and, respectively, EGFR as a main factor in ERK phosphorylation [55,56]. Taken together, we observed a correlation between the activation of AKT and ERK as routes of downstream signaling that is prominently triggered by the respective ratios of EGFR and ERBB3 receptors in the cell lines tested.

Having validated our initial hypothesis, we next wanted to address three main questions: (i) Are there other variables that determine downstream signaling and cellular response on top of the receptor composition? If true, (ii) what other factors are needed to explain the response? Furthermore, (iii), can we establish a unified model of ERBB signaling for the luminal, TNBC and HER2-positive cell-line systems that is able to predict the effect of ligand stimulation and drug treatments for the three breast cancer subtypes?

### 3.2. Development of a Mechanistic Model of the ERBB Signaling Network

To explore the differential downstream activation of breast cancer subtypes, we established an ODE model describing the ERBB signaling network as displayed in Figure 2. Reaction kinetics were modeled by the law of mass-action according to chemical reaction network theory [57]. This signaling model represents a mechanistic description of growth-factor induced signal transduction at the receptor module towards the two branches of the downstream module, i.e., PI3K as well as MAPK signaling.

Upon ligand binding, several ERBB homo- and heterodimers form, leading to the trans-activation of receptors and to the induction of downstream signaling [2]. Our model simplified these two steps in one, i.e., the formation of signaling-competent active dimers, considering EGFR homodimers as well as well as EGFR–ERBB2, EGFR–ERBB3 and ERBB2–ERBB3 heterodimers. Due to the distinct binding affinities the four ligands have to the different ERBB receptors, each ligand was modeled to have individual dimerization rates.

Once activated, the receptor dimers induce phosphorylation and thereby the activation of the MAPK and PI3K signaling pathways, which ultimately converges in the activation of P70S6 kinase in our experimental system [58]. The mathematical model was built with the objective to capture the signaling dynamics of key components within the signaling network and yield well defined parameter estimates and predictions. Therefore, its size and complexity were tailored to the complexity of the available data. In other words, only reactions were included that were essential for the description of the data [59].

Using the complete data set, the model was first implemented for the luminal cell lines MCF7 and T47D. The time-resolved total and phosphoproteomics data of EGFR, ERBB2, ERBB3, c-RAF, MEK, ERK, AKT and P70S6K proteins are complemented by the absolute protein abundances taken from Niepel et al. [54] and used for model calibration. The final data set comprised 3600 data points from 24 conditions per cell line, merged from three biological replicates each. To evaluate drug and ligand effects individually, the time course experiment was divided into two experimental phases.

In phase I, the inhibition phase, we assessed drug-induced changes independent of any ligand, while in phase II, we addressed the dynamics of the ligand-induced protein activation in unperturbed and in drug-perturbed conditions (Figure 3a). Samples were taken at ten time points during the two phases and analyzed with targeted proteomics using reverse phase protein array (RPPA) technology.

Figure 3b displays the differential response of MCF7 cells to the four ligands. Corresponding data for T47D are shown in Appendix A. An instant increase in phosphorylation is visible for all ligands, which was only slightly delayed in the most downstream protein of the cascade tested, P70S6K. This activation was followed by a ligand-specific decrease of phosphorylation that varied between targets. Consistent with the high ratio of ERBB3 in this cell line, NRG1 induced the strongest activation of all downstream proteins, while the response of the three EGFR ligands was smaller particularly on AKT. These results are consistent with the dominant role of ERBB3 in the activation of the PI3K pathway [59,60].

### 3.3. Impact of ERBB Inhibitors

Ligands induce a conformational change, dimerization and activation of their respective receptors, whereas our drugs of interest either inhibit receptor dimerization (pertuzumab), impede receptor activation at the level of the extracellular domain (cetuximab, trastuzumab, lumretuzumab) or block ATP-binding in the kinase domain (erlotinib). We implemented this inhibition in the mathematical model by multiplying the dimer formation and activation rate with an inhibition term $\frac{1}{1+k_{drug}}$. Thereby, the parameter $k_{drug}$ determines the strength of inhibition per drug.

As shown in Figure 3c, this implementation enabled the model to capture the diverging activation dynamics of downstream proteins under the different drug treatments. Drugs either reduced the total phosphorylation level of affected phospho-proteins and/or sharpened their signaling peak. NRG1 induced the strongest activation in the luminal cell lines (Figure 3b).

Drug effects in combination with this ligand for MCF7 are displayed in Figure 3c, while the inhibitory impact of the drugs under stimulation with EGF, TGFa and BTC are depicted in Appendix A for MCF7 and in Appendix A for all ligands in T47D. The efficacy of the different drugs was dependent on the respective ligand applied. As expected, lumretuzumab had the strongest inhibitory effect when cells were stimulated with NRG1, followed by pertuzumab, which also led to a remarkable decrease in protein activation.

Trastuzumab and erlotinib only showed a minor inhibitory effect in combination with this ligand, while cetuximab did not have any. However, this picture turned when cells were stimulated with one of the EGFR ligands. Erlotinib and cetuximab led to a strong reduction of protein phosphorylation, in particular, upon stimulation with EGF or TGFa, while the other drugs had only minor effects (Appendix A). In addition to the described inhibitory effects, the following additional drug activities were identified.

(i) Trastuzumab as well as pertuzumab induced ligand-independent phosphorylation of ERBB2 at Tyr^877^ and Tyr^1221^/Tyr^1222^, of EGFR at Tyr^1086^ and to a lesser extent also of ERBB3 at Tyr^1289^ (Appendix A). This effect was not observed for other ERBB tyrosine residues, such as Tyr^1148^ of pEGFR, Tyr^1197^ or Tyr^1222^ of ERBB3.

The observed phosphorylation pattern was reflected in the phosphorylation at Tyr^416^ of the protein kinase SRC. This suggests that these antibodies might trigger phosphorylation of ERBB receptors involving the Src kinase. This drug-induced phosphorylation was stronger in the T47D cell line compared to MCF7 and more pronounced when incubated with trastuzumab as compared to pertuzumab (Appendix A).

Cetuximab did not induce any receptor phosphorylation suggesting that this was specific for ERBB2 and the respective targeting drugs. However, none of these activated receptors induced downstream MAPK or PI3K signaling (Figure 3c). We thus implemented this apparent ligand independent receptor phosphorylation in our model by introducing additional model states for phosphorylated yet inactive receptors (Figure 2).

(ii) Lumretuzumab and pertuzumab both led to a strong inhibition of ERBB3 phosphorylation (Figure 3d, middle panel). In addition to this inhibitory effect on ERBB3 activity, lumretuzumab induced strong downregulation of the total ERBB3 protein level (Figure 3d, right panel), which was further enhanced in presence of NRG1. No similar effect was observed upon incubation with any other drugs. Thus, lumretuzumab reduces not only ERBB3 activity, but also induces rapid degradation of ERBB3.

Taken together, the signaling model we developed for MCF7 and T47D cell lines was able to capture the dynamics of proteins involved in the MAPK and the PI3K pathways, covering a range of ligand–drug combinations. It comprises 22 model states, 44 reactions and 188 parameters, whereof 70 parameters are dynamic, describing kinetic rates and protein abundances, and 118 are observation-related. The full set of reactions and corresponding observables are listed in Appendix A, estimated parameter values along with their 95% confidence intervals, computed by the profile likelihood method [43], are displayed in Appendix A.

### 3.4. Predicting Drug Effects on Ligand-Dependent Proliferation

Next, the calibrated signaling model was extended with a proliferation model as a functional readout to predict the proliferative capacity of cell lines in presence of the different drugs and ligands. Applying linear regression, this proliferation model should establish a link between intracellular signaling dynamics and a cancer-relevant phenotype. To this end, we first simulated response dynamics of single model states and then applied these to generate signaling features, such as the area under the curve (AUC), the slopes and the peak height of states of interest.

The selection of states used for the generation of the feature set was made up of pAKT, pERK, pP70S6K and the active dimers, which were previously shown to have predictive power [24]. These features were then used to describe the condition-specific proliferation response via the linear regression model. To calibrate this model, a proliferation screen was carried out with MCF7 and T47D cell lines, where we initially tested experimental conditions with single drug treatments.

Cells were treated with the same drug-ligand combinations that had been used to obtain the original data set. Then, a subset of signaling features was selected from the complete feature set, based on their prediction ability as determined by L_1_ regularization. Our analysis showed that the AUC of pAKT combined with the AUC of pP70S6K had the highest predictive power. Thus, these features served as predictor variables to describe the phenotypic readout, i.e., cell proliferation.

The predictive power of all signaling features tested is summarized in Appendix A. In general, the peak height showed the best performance for the active dimers, while the area under the curve was the most predictive feature for downstream signaling components. A schematic overview of the proliferation modeling process is depicted in Figure 4a, and a detailed summary of regression parameters is given in Appendix A.

We then wanted to test if the resulting proliferation model would correctly predict treatment efficacy. To this end, we generated growth predictions for 24 combination treatments. The model suggested the co-treatment of pertuzumab with erlotinib as the overall most efficient treatment in the luminal A cells (Figure 4b). To validate these predictions and test the predictive power of the model, validation experiments were performed for three selected drug combinations per ligand, which showed the expected effects on cell growth.

Model predictions and validation data were in good agreement, featuring only a minor increase in the mean estimated error of 0.005, representing 3% of total data spread, compared to the training data set of single treatments (Figure 4c). These results suggest that the developed proliferation model is also applicable for the prediction of cell growth in response to single and combinatorial drug treatments. In conclusion of this part of the study, the complementation of the dynamic signaling model with the proliferation model allowed to bridge from the molecular activation to the phenotypic level, in response to various combinations of drugs and ligands.

### 3.5. Model-Based Analysis of Cell Line-Specific Activation Kinetics beyond the Luminal Subtype

We next wanted to assess whether the identified network kinetics are luminal cell-line-specific or if the model could be adapted also to other breast cancer subtypes. To this end, we examined the ligand-induced phosphorylation of MAPK and PI3K pathway proteins under drug influence in TNBC MDA-MB-231 as well as HER2-positive SKBR3 cells.

Time-resolved phosphoproteomics data was generated for these two cell lines applying the same experimental conditions that we previously used with the luminal cell lines, except that we induced signaling in the SKBR3 cells with only EGF and NRG1 and that we used pertuzumab, trastuzumab and lumretuzumab as inhibitors. To identify key functions that are differentially regulated between these cell lines, differences in parameters of the established signal transduction model were analyzed based on L_1_ regularization as previously described [61].

In total, 3600 data points were collected for the MDA-MB-231 cell line and another 680 data points for SKBR3, to quantify dynamic changes under the experimental conditions tested. These data were then used to calibrate the additional cell-specific model parameters along with their protein abundances, which were fitted without L_1_ prior (Appendix A). All other dynamic parameters were fixed to the estimated values obtained from the luminal cell lines. L_1_ analysis revealed relevant differences for the other cell lines, however, only for certain parameters.

As shown in Figure 5a, in particular, the basal phosphorylation rate of MEK and the basal activation ratio of c-RAF, a parameter describing the ratio of activation and deactivation of c-RAF, were different in MDA-MB-231 as well as in SKBR3 compared to the luminal cell lines. In addition, the deactivation rate of P70S6K was increased in both cell lines, while all other dynamic parameters indicated only minor impacts.

Accounting for these identified parameter differences, the resulting model trajectories were fully in agreement with the experimental data (Figure 5b, Appendix A) thus also confirming the successful adaptation of the signaling model to the cell lines representing two further breast cancer subtypes.

Similar to the T47D and MCF7 luminal cell lines, we next wanted to determine if the therapeutic ERBB2-binding antibodies could induce phosphorylation of the ERBB-receptors in these cell lines of other breast cancer subtypes as well. HER2-positive SKBR3 is dominated by the ERBB2 receptor, while the TNBC cell line MDA-MB-231 mostly expresses EGFR.

Accordingly, we observed even stronger phosphorylation of ERBB2 in SKBR3 compared to the luminal cell lines. Interestingly, the autophosphorylation sites Tyr^1221^/Tyr^1222^ were stronger phosphorylated in this cell line compared to Tyr^877^ (Appendix A). In contrast, none of the RTKs was phosphorylated in the MDA-MB-231, which is consistent with the low expression of ERBB2 in this cell line. Phosphorylation of Src kinase at Tyr^416^ followed the same pattern is SKBR3 as we had observed in T47D cells.

### 3.6. Mutation States Impact the Dynamics of Signaling

Consistent with the results obtained in the L_1_ analysis, MDA-MB-231 as well as SKBR3 showed low basal activity of c-RAF in the first 60 min without ligand (Figure 5b). However, while this was accompanied by a similarly low basal activation of MEK in the HER2-positive SKBR3, an increased activation of MEK phosphorylation was apparent in MDA-MB-231. We suspected that the particular mutational states of the cell lines in our study might impact this differential behavior.

Furthermore, two particular molecular alterations in MDA-MB-231 might lead to these opposing activation kinetics. This cell line carries an activating KRAS mutation (G13D) as well as an activating b-RAF mutation (G464V) [63]. Given the position of MEK directly downstream of the RAS and RAF proteins in the signaling cascade, a constitutively active b-RAF protein could explain the elevated activity of MEK, which would be independent of c-RAF.

In addition, this specific mutation in b-RAF has previously been reported to trigger feedback inhibition of RAS and c-RAF by ERK, which should result in a decrease in the basal activation ratio of c-RAF [64,65]. To test the relevance of this mechanism in our cell system, we quantified the c-RAF phosphorylation at positions Ser^289^/Ser^296^/Ser^301^.

These residues are phosphorylated by ERK and inhibit the interaction between RAS and c-RAF thereby leading to the inactivation of c-RAF as part of a negative feedback loop [62]. Matching our expectations, basal phosphorylation at this inactivation site was elevated in MDA-MB-231 compared to the other cell lines tested (Figure 5c). This was accompanied by a reduced response to ligand activation thereby confirming the presumed mechanism. In contrast, SKBR3 are wild type for the genes in the MAPK signaling pathway.

In this cell line, a strong preference towards PI3K signaling was observed, which was reflected by the low activation rates of c-RAF and MEK as well as an increase in the activation rate of P70S6K by AKT (Figure 5a), compared to the other cell lines. Moreover, the elevated basal phosphorylation of AKT and P70S6K in SKBR3 cells underlines the dominant role the PI3K pathway has in this cell system (Figure 5b), which is driven by the amplification of the ERBB2 gene locus and high-level expression of ERBB2.

### 3.7. Flux-Analysis of MAPK versus AKT Signaling

Finally, we analyzed the cell-line-specific activation of signaling branches. Along these lines, we analyzed the activation fluxes of the P70S6 kinase via either pERK or pAKT and then calculated their respective log10 ratios (Figure 6a). As the MAPK and PI3K pathway branches converge at P70S6K, we considered this ratio as an estimate of the relative activities of these two pathways.

A positive score would indicate dominance of the MAPK pathway in the activation of P70S6K over PI3K signaling and vice versa. Signaling via AKT indeed dominated the steady state in the luminal cell lines, which can be explained with activating the hot-spot mutations in *PIK3CA* present in MCF7 and T47D. However, in these cell lines, the MAPK signaling gained relevance upon stimulation of EGFR, leading to a transient switch towards MAPK-regulated activation of P70S6K. Yet, this impact was short, and the predominant activation via AKT quickly reestablished.

This overshoot behavior reflects the transient dynamics of ERK phosphorylation—particularly when the EGFR was activated by EGF. The dynamics of ERK-activation were longer-lasting with the other two EGFR-ligands tested, TGFa and BTC, both leading to a shortly extended activation of P70S6K by ERK. The transient nature of P70S6K activation via EGFR was in contrast to the longer lasting activation via ERBB3 particularly in MCF7, which was clear upon activation with NRG1 (see also Figure 3b).

In sharp contrast to the two luminal cell lines, P70S6K was mostly activated by MAPK signaling in MDA-MB-231 cells (Figure 6a). This was independent of the respective ligands used to activate the system, as all ligands induced considerable phosphorylation of MAPK proteins and the most pronounced for MEK (see also Figure 5b and Appendix A). In line with the results described above, this is consistent with the high prevalence of EGFR in this cell line, in combination with an activating mutation in *b-RAF*.

Along the same lines, stimulation with NRG1 did not lead to a sizeable activation of any downstream signaling proteins and thus did not affect the P70S6K activation flux ratio (see also Appendix A). The levels of ERBB3—the receptor for NRG1—are thus likely too low to be relevant in this system. Different from MDA-MB-231, the main signaling flow in the HER2-positive SKBR3 cell line was driven by amplification and the high-level expression of the ERBB2 protein, leading to P70S6K activation mostly via the PI3K pathway.

The dominance of PI3K signaling is so strong in SKBR3 that, even with activation of EGFR and downstream MAPK signaling, a balanced ratio of P70S6K activation was not reached for the MAPK and PI3K pathways. ERBB2 is the preferred binding partner of ERBB3, which has six p85 binding sites in its intracellular domain and thus preferentially induces AKT signaling [66,67]. Even though the levels of EGFR are somewhat higher than those of ERBB3 in SKBR3, still the ERBB2–ERBB3 heterodimers appeared to strongly outweigh the MAPK signaling that would be induced by EGFR–ERBB2 heterodimers. ERBB2 and downstream PI3K signaling are thus clearly the drivers of cancer properties in this cell line, matching the clinical relevance of ERBB2/HER2 as the therapeutic target in this subtype [68].

Based on the experimental data as well as the cell-line-specific parameters we identified in modeling, we propose specific major activation routes for the different cell lines Figure 6b. These routes were strongly impacted by specific protein abundances and mutation states, such as negative feedback inhibition in MDA-MB-231 cells, as most kinetic rates were identical between the cell lines. Still, our model correctly describes the respective dominating signaling fluxes within all four cell lines in our study.

Thus, we established a unified mathematical model for ERBB and downstream MAPK as well as PI3K signaling, which required only a few biologically relevant adaptations to correctly describe the molecular responses of luminal A T47D and MCF7, TNBC MDA-MB-231 and HER2-positive SKBR3 cell lines upon stimulation with several ligands as well as perturbations with drugs targeting EGFR, ERBB2 and ERBB3.

## 4. Discussion

The ERBB signaling network plays a crucial role in the regulation of cellular growth, migration and survival, which are all critical phenotypes in health and disease [2,9]. Although various cancer entities share the same molecular network topology of signaling pathways, they exhibit differential phenotypic responses to ligands and drugs interacting with particular network components. In cancer, these responses are predominantly triggered by the mutation states of signaling proteins.

For example, activating mutations in *EGFR* are frequent in relapsed lung adenocarcinoma, *BRAF* is often mutated in melanoma, and *PIK3CA* is often mutated in various entities, including luminal breast cancer. Beyond single nucleotide variants, the *ERBB2* gene locus is amplified and a diagnostic marker in several entities, including HER2-positive breast cancer. Together, these factors affect tumor physiology. However, the complex interplay of signaling pathways and how these trigger cancer phenotypes in response to different inputs is not fully understood.

Here, we established a mathematical model that mechanistically describes the activation dynamics of ERBB proteins and of two prominent downstream pathways, RAF-MEK-ERK and PI3K-AKT. The amounts and phosphorylation states of central components within these pathways were quantified with high temporal resolution to capture dynamic changes in their respective abundance and activities.

To perturb and thereby constrain the proliferation signaling of the cell lines in our study, five drugs targeting the ERBB receptor family were analyzed as single agents as well as in combination, always in the presence of different ligands. The drugs specifically targeted the EGFR, ERBB2 and ERBB3 receptor tyrosine kinases, while ligands were selected to either activate EGFR or ERBB3. The ODE model was then calibrated on the experimental data of single drug treatments, initially from the two luminal breast cancer cell lines, MCF7 and T47D.

To establish a link between pathway activation and cell proliferation as our phenotypic readout, we combined this mechanism-based signaling model with a linear regression model. Signaling features, such as the area under the curve of single activated proteins, were simulated by our signaling model and served as input for the regression model. The regression model was then calibrated on experimental data, which we collected in a proliferation screen depicting the proliferative response of MCF7 and T47D cells under the respective ligand–drug treatments.

The coupling of signaling and proliferation models, both calibrated on single drug treatments, made it possible to reliably predict phenotypic responses also to co-treatments imposed at the receptor level. Thereby, our combined model could be validated. This approach of linking an ODE signaling model with a proliferation regression model enabled us to find a balance between the mechanistic complexity and the comprehensibility of the model.

The validity of this implementation is supported by the fact that the applied drugs directly target signaling components, i.e., the growth factor receptors in our study [16]. Thus, we can now describe drug effects in mechanistic detail while inferring the proliferative behavior of cells from signaling features using regression modeling.

Having analyzed two different luminal cell lines, we wanted to know if cell-line-specific differences in the activation of downstream signaling branches needed to be considered in the model. Regarding the initial model calibration on the luminal cell lines MCF7 and T47D, model parameters could indeed be implemented as identical between these two cell lines with the exception of the respective protein abundances as well as the *PIK3CA* mutation status.

These were estimated separately for each cell line [69]. This is in line with the similar receptor composition in both cell lines that are dominated by ERBB3 and ERBB2 and distinct activating mutations present in PIK3CA, i.e., E545K in MCF7 and H1047R in T47D. Hot-spot mutations in PIK3CA were previously reported to uncouple ERBB2–ERBB3 signaling from PI3K activity thereby rendering mutated cell lines insensitive to trastuzumab treatment [70].

Therefore, our data and the model suggested that the H1047R mutation in the kinase domain has a stronger effect than the E545K mutation in the helical domain. This observation is in line with the increased basal activation of pAKT in the T47D cell line. These properties of the two luminal cell lines lead to a dominance of the PI3K-AKT signaling branch also in ERBB-downstream signaling. *PIK3CA* is the most frequently mutated gene in luminal breast cancer and the same hotspot mutations present in MCF7 and T47D have been found in 45% of patient tumors, underlining the clinical relevance of the model systems in our study [71].

In order to assess whether our model was also applicable to TNBC, MDA-MB-231 cells were tested in the same experimental setup as used with the luminal cell lines. L_1_ regularization was applied to analyze whether, and if so, which model parameters were necessary to be adjusted, for this cell line to compensate for cell-line-specific differences in the activation of downstream signaling branches. In this process, both inactive c-RAF as well as increased basal activation of MEK were identified as major distinguishing factors between MDA-MB-231 and the luminal cell lines.

Indeed, the phosphorylation of MEK and ERK was prolonged and elevated even under steady state conditions in MDA-MB-231, which could be due to an activating b-RAF mutation that is present in this cell line [63]. This b-RAF mutation maintains the protein in an active conformation and likely leads to the activation of MEK as well as to a negative feedback loop involving the deactivation of c-RAF [64].

These molecular changes combined with a strong dominance of EGFR in this TNBC cell line, led to a shift between pathway branches from PI3K signaling towards MAPK signaling, standing in sharp contrast to the pathway regulation in the luminal cell lines. Taking these factors into account, our mathematical model then required few further adjustments regarding the receptor dimerization rates to reliably fit the experimental data. Hence, the same model could be applied to luminal and TNBC breast cancer cell lines, which have distinct receptor compositions, mutation states and growth properties and thus going beyond previous modeling approaches [31].

We then wanted to know whether the model could correctly predict the signaling dynamics also for the HER2-positive subtype of breast cancer. To this end, we employed the SKBR3 cell line as a suitable experimental system. This cell line has a high ratio of ERBB2 over all other members of the ERBB family. L_1_ regularization identified the basal activation ratio of c-RAF as the most important parameter necessary to be adjusted. However, while the basal MEK phosphorylation was enhanced in the MDA-MB-231 cell line, this parameter was reduced in SKBR3.

Hence, the MAPK signaling pathway seems to be less relevant in SKBR3 as compared to the luminal cell lines MCF7 and T47D, whereas the PI3K pathway strongly dominates signaling downstream of the ERBB receptor level. However, even though the SKBR3 cells appear to be addicted to PI3K signaling, adjustment of our model for the highly biased expression of the signaling components as well as for the other identified cell-line-specific parameters, allowed to precisely describe the experimental data. Hence, our mathematical model is able to comprehensively reflect ERBB signaling towards RAF-MEK-ERK as well as PI3K-AKT signaling in three subtypes of breast cancer.

EGFR and other RTKs are flexible in structure, and the binding of growth factors increases their probability of adopting a state, which is favorable for dimerization [72,73], transphosphorylation and activity. For EGFR, it has been reported that cetuximab binds domain III of the extracellular domain of EGFR thereby reducing the conformational flexibility and fixing the protein in a compact conformation that does not allow dimerization [74].

In contrast, the presence of an extended tertiary structure is an intrinsic feature of ERBB2, which does not require structural changes during the formation of active heterodimers [72]. However, the lack of flexibility of ERBB2 also prevents formation of a binding pocket that would be required to home the dimerization arm of the binding partner thus requiring a growth factor-bound binding partner [75]. Hence, no dimerization and activation of receptors seems to occur in the absence of growth factors.

We observed the phosphorylation of ERBB2 at position Tyr^877^ and, to a lesser extent, at Tyr^1221^/Tyr^1222^ upon incubation with either trastuzumab or pertuzumab, which was more prominent with trastuzumab compared to pertuzumab and in T47D compared to MCF7 cell lines. A similar effect was not observed with cetuximab suggesting that the phosphorylation related to the specific antibodies and/or ERBB2. In HER2-positive SKBR3, Tyr^1221^/Tyr^1222^ were found even more strongly phosphorylated while MDA-MB-231 cells appeared to not be affected by any of the therapeutic antibodies. The latter observation might be related to the predominance of EGFR expression in MDA-MB-231.

Tyr^877^ is phosphorylated by Src [76] while Tyr^1221^ and Tyr^1222^ are autophosphorylated by ERBB2 [77] suggesting that Src could be recruited to ERBB2 upon binding of either trastuzumab or pertuzumab and prime the RTK for some activation. Furthermore, Src kinase was activated upon incubation of the cells with trastuzumab or pertuzumab at similar ratios as the RTKs. Phosphorylation and activation of ERBB2 commonly involves heterodimerization with EGFR or ERBB3. We also tested phosphosites indicative of activities in these other two receptors.

Similar to the phosphorylation of ERBB2, we observed phosphorylation also of EGFR Tyr^1086^, however, not at Tyr^1148^. Both these sites are autophosphorylated by EGFR [78]. However, Tyr^1086^ is also a target of Src kinase [79]. Hence, the phosphorylation of ERBB2 and interestingly, also of EGFR might be triggered by Src and not via ERBB2 (alone).

Along the lines of activation of ERBB2, phosphorylation of EGFR was not changed by cetuximab in any of the cell lines tested, while we also observed phosphorylation of ERBB3 with trastuzumab and pertuzumab, this was much less compared to ERBB2 or EGFR in all tested lines. This was unexpected as the ERBB3-receptor is even the most strongly expressed member of the ERBB-family in MCF7.

The phosphorylation of all kinases was strongest in SKBR3, followed by T47D and MCF7 cells. This order reflects the absolute numbers and receptor-ratios in these cell lines regarding expression of ERBB2. Elevated phosphorylation of EGFR and ERBB2 in response to treatment with trastuzumab has been reported before in ERBB2-overexpressing cell lines [80]. Here, we show that this occurs even when EGFR and ERBB2 levels are rather low. It needs to be assessed if this is direct or, for example, via Src.

Dokmanovic et al. investigated the impact of trastuzumab at the level of receptor phosphorylation and stability [80]. Trastuzumab was found to downregulate the total protein levels of ERBB2 and ERBB3. However, the observations were not consistent for the two cell line systems used in that study. ERBB2 was stable in SKBR3, however, downregulated in BT474, a HER2-positive luminal B cell line, within 60 min of incubation with trastuzumab.

Hence, the effects of trastuzumab on receptor stability seem to be cell line-dependent. A reduction in ERBB3 protein levels was observed after several (i.e., 3–5) days of treatment with trastuzumab. Here, we found particularly lumretuzumab to reduce total ERBB3 levels within minutes and even without any ligand.

In the time-scale of our measurements, the levels of ERBB3 were independent of drugs other than lumretuzumab, including erlotinib, and were then only and similarly reduced after stimulation with NRG1. Therefore, the dynamics of receptor downregulation seem to depend on the specific antibodies, the receptors and their abundance as well as on the specific growth factors.

Different growth factors and downstream pathways are commonly associated with particular phenotypic responses, such as proliferation, migration and survival [81,82,83]. The model we developed captures the ligand-dependent formation of different receptor homo- and heterodimers and their contribution to the activation of downstream signaling, with focus on the ERBB family of receptor tyrosine kinases.

Previous modeling studies have followed a similar idea [24,31]. However, in contrast to these other studies our model includes the mechanistic impact of several kinase inhibitors in combination with different growth factors and their effects on signal transduction as well as on a cancer-relevant phenotype. Notably, in the in vivo setting, cells mostly face a combination of different ligands.

These originate from autocrine or paracrine signaling loops also involving the tumor microenvironment [84,85]. This scenario needs to be taken into consideration particularly in the clinical setting. In addition, multimodal effects of antibodies in vivo, such as antibody-dependent cell-mediated cytotoxicity (ADCC), are not captured by our in vitro model [86,87].

Furthermore, our implementation contains simplifications in terms of regarded receptors. It has been shown, for example, that the insulin-like growth factor receptor (IGF-R) and the hepatocyte growth factor receptor (cMET) mediate a variety of cellular processes in breast cancer cells as well [88,89]. It remains to be determined how these other receptors impinge on this network as well as on its wiring.

Exploiting our experimental data as well as the mathematical model we were finally out to uncover interactions between ERBB signaling and cell proliferation. Our model implied that co-treatment of luminal cell lines with pertuzumab and erlotinib should have the strongest effect on proliferation. Furthermore, we confirm this in experiments. It should be noted, however, that the observed effects were mild in the tested cell lines.

This is in line with the physiology of luminal breast cancer, which is driven by the estrogen receptor much more than by ERBB receptor-signaling. In addition, the uncoupling of PI3K signaling in these cell lines likely supports this phenotype due to the mutated PIK3CA [70]. Still, the ERBB and downstream MAPK and PI3K signaling network has received attention also in the luminal subtypes. Previous studies reported that the ERBB-network is induced in conditions when tumors have acquired resistance to endocrine therapies [90,91].

Endocrine resistance comes also with an upregulation of ERBB-proteins and, according to our model, the receptor composition is likely one major determinant of ERBB and downstream pathway activities on tumor physiology then. Along the same lines, the PIK3CA-inhibitor alpelisib has been approved for treatment of hormone receptor positive, HER2-negative metastatic breast cancer further supporting the impact PI3K-signaling has in the luminal subtypes once endocrine resistance has kicked in [92] and supports the concept of targeting the ERBB signaling network also in HER2-low cancer conditions [93].

## 5. Conclusions

Here, we developed a modeling strategy that combines the molecular ERBB signaling network with the proliferation phenotype. We determined the relevant variables that affect downstream MAPK and PI3K signaling and presented a unified model that enabled us to assess the effect of single and combination treatments and to predict the resulting proliferative behavior in cell lines representing three subtypes of breast cancer. At the same time, the model provided insights into the regulation of the downstream pathway activation and pointed at a shift in activated signaling branches between breast cancer cells of different subtypes.

The weighting of cell-type-specific characteristics, which enabled us to analyze this activation of the individual signaling branches and construct one model for all cell lines, represents the major novelty of this work. The complexity of the biological systems was analyzed as well as the responses to drugs and ligands that we could infer, highlight the potential of mathematical modeling for the prediction of signaling readouts and hints at mechanisms in the underlying biology. Our study further supports the concept of anti-HER therapies in HER2-low breast cancer.

## Figures and Tables

**Figure 1 cancers-14-02379-f001:**
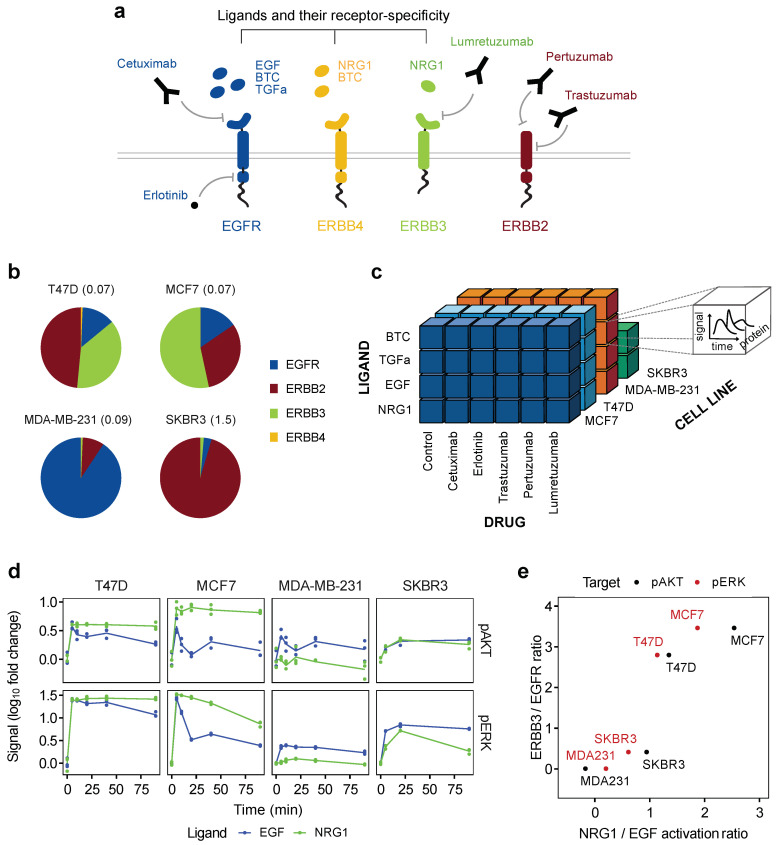
The ERBB receptor composition in relation to the downstream signaling response. (**a**) Overview of ERBB receptors and their interactions with specific ligands and antagonistic inhibitors. ERBB2 is missing a ligand-binding domain, whereas ERBB3 is lacking a kinase domain. (**b**) The ERBB receptor composition of the four analyzed cell lines. Numbers in parentheses indicate the total amount of ERBB receptors in pg/cell and were taken from Niepel et al. [54]. (**c**) The experimental data set comprises dynamic protein measurements of four different ligand stimulations combined with five drug pre-treatments and a control for four cell lines. The resulting experimental conditions per cell line are depicted as a cube each. In every condition, dynamic alterations of several proteins and phospho-proteins were quantitatively assessed. (**d**) Time-resolved activation dynamics of AKT and ERK were quantified from RPPA measurements for T47D, MCF7, MDA-MB-231 and SKBR3 cells. The three biological replicates are displayed as dots, and lines represent linear interpolations. (**e**) Linear correlation of the activation ratio of downstream signaling components by NRG1 vs. EGF and the ERBB3 vs. EGFR expression ratio. The activation of AKT and ERK was assessed using the area under the curve (AUC) of each stimulus shown in panel (**e**).

**Figure 2 cancers-14-02379-f002:**
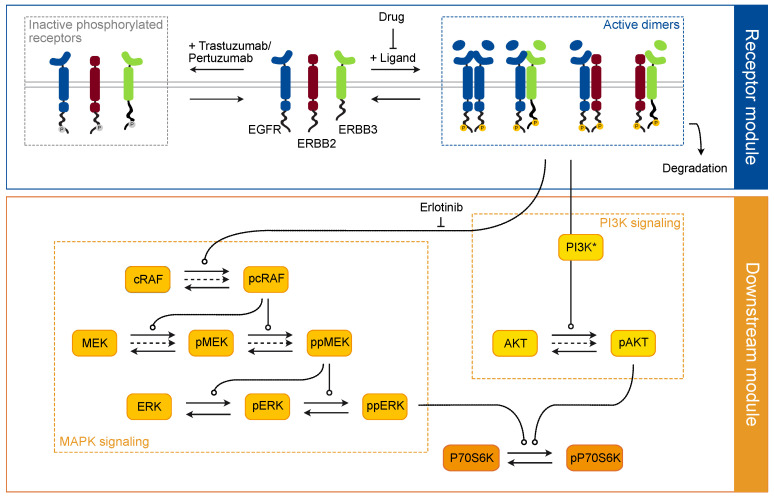
The structure of the mathematical signaling model. Receptor module: The ERBB receptors EGFR, ERBB2 and ERBB3 form several homo- and hetero-dimers upon ligand binding, which lead to their trans-phosphorylation and thereby activation (dotted blue box ’Active dimers’) and eventually to their degradation. Pertuzumab and trastuzumab exhibit an additional function, inducing the ligand-independent phosphorylation of receptors (dotted grey box ’Inactive phosphorylated receptors’). Downstream module: Activated receptors signal downstream through the MAPK and PI3K cascades (the respective dotted yellow boxes) ultimately converging in the phosphorylation of P70S6K. Drugs inhibit the formation of active dimers and thus downstream signaling. Reactions are represented by arrows, dashed lines indicate basal reaction fluxes. The asterisk indicates a mutation in PIK3CA.

**Figure 3 cancers-14-02379-f003:**
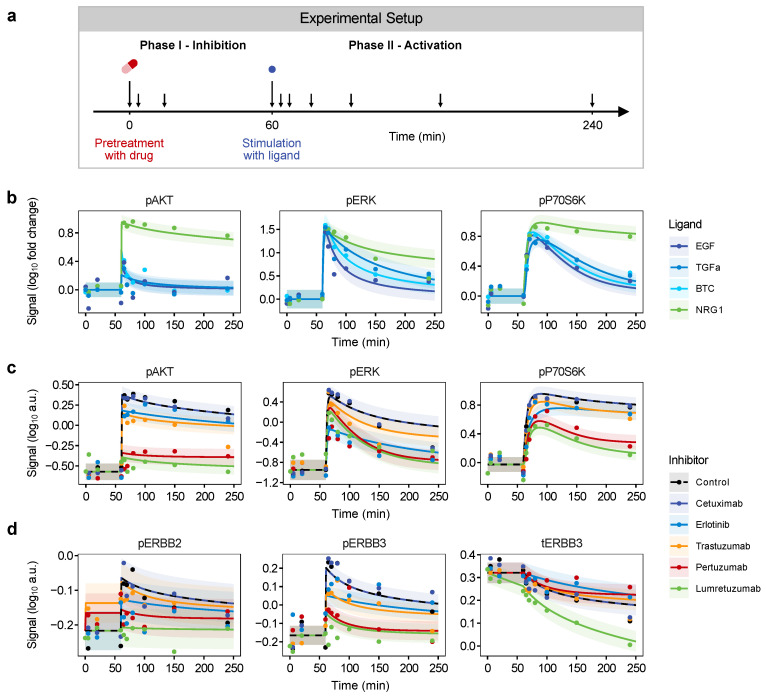
Model calibration on time-resolved quantitative reverse phase protein array (RPPA) data. (**a**) RPPA data generation. Cells were pretreated with drugs for 60 min (phase I) and then stimulated with ligands for up to another 180 min (phase II). Samples were taken at ten time points between 0 and 240 min, as indicated by arrows. (**b**–**d**) RPPA measurements and model fits of MCF7 cells. (**b**) The cellular response to the four ligands is depicted for the three major downstream proteins pAKT, pERK and pP70S6K. (**c**) The impact of the five drugs under NRG1 stimulation is shown for the same downstream proteins (**d**) as well as for pERBB2, pERBB3 and total ERBB3 (tERBB3). All data represents the mean of biological replicates (n = 3) and is displayed as dots. Trajectories of the model fit are indicated as solid or dashed lines along with shaded error bands corresponding to one standard deviation. In total, the data of eight proteins and four ligands in MCF7 as well as T47D cells were used for the initial model calibration. Corresponding data and model trajectories for T47D are depicted in Appendix A.

**Figure 4 cancers-14-02379-f004:**
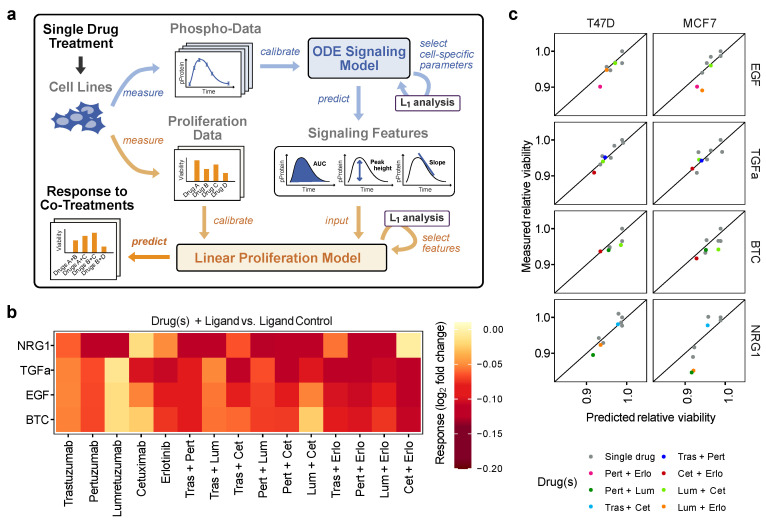
Model-based proliferation predictions. (**a**) Schematic overview of the proliferation prediction process. Signaling features were simulated based on the calibrated ODE model and used as input for the linear proliferation model to predict condition-specific proliferation responses. Single treatments were used for model calibration to predict the response to co-treatments. (**b**) Proliferation predictions of MCF7 cells pre-treated with a single or a combination of drugs, followed by a stimulation with the indicated ligand. Data are displayed as log2 fold change relative to the ligand-only control. (**c**) Cell viability data were experimentally collected and used to assess the predictive power of the model for co-treatment predictions. Single treatments used for model calibration are shown in grey, while validation co-treatments are colored. Drugs are abbreviated for co-treatment denotation: Cet—cetuximab, Erlo—erlotinib, Tras—trastuzumab, Pert—pertuzumab and Lum—lumretuzumab. Data points are displayed as the fold change relative to the ligand control.

**Figure 5 cancers-14-02379-f005:**
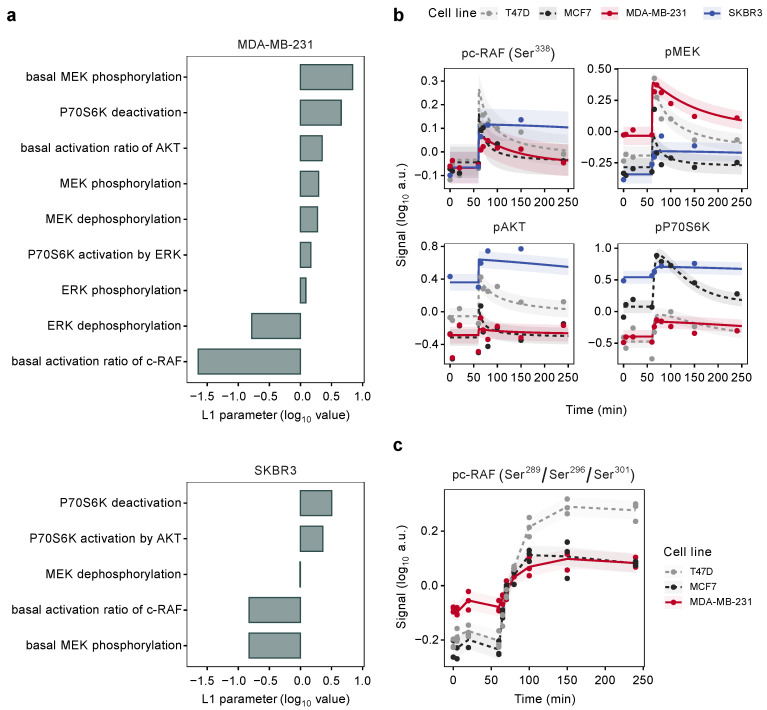
L_1_ regularization identifies cell-line-specific parameters. (**a**) Regularization-dependent parameter differences $\log_{10}\left(r_{i}\right)$ for the downstream module of MDA-MB-231 and SKBR3 cells. Positive values indicate elevated reaction rates relative to the luminal cell lines, while negative values show a decrease. AIC statistics were used to determine the optimal regularization strength for the analysis. (**b**) Model fits and experimental data (mean of 3 biological replicates) for the activating c-RAF, MEK, AKT and P70S6K sites in all four cell lines upon EGF stimulation. (**c**) Experimental testing of c-RAF phosphorylation (RPPA, n = 3) at positions Ser^289^/Ser^296^/Ser^301^, which are phosphorylated by ERK in a negative feedback mechanism to downregulate MAPK signaling [62].

**Figure 6 cancers-14-02379-f006:**
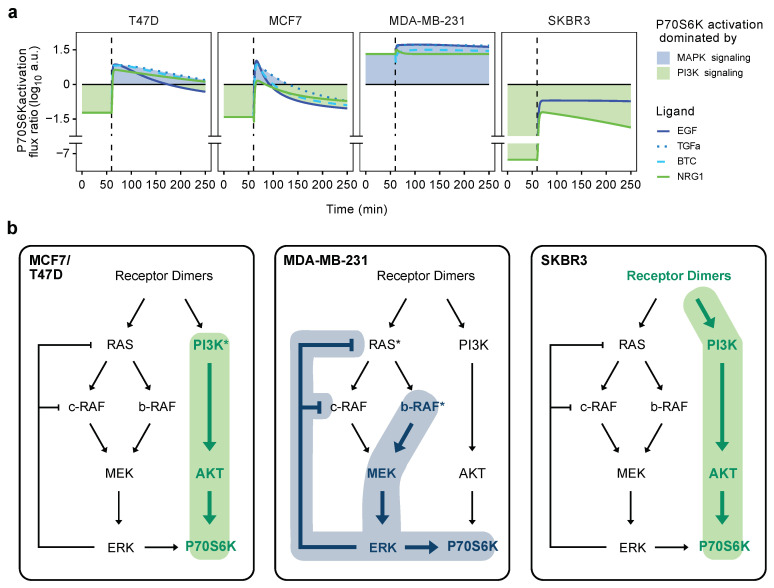
Signaling fluxes strongly differ between breast cancer cell lines. (**a**) The activation of the P70S6 kinase by MAPK and PI3K signaling pathways is depicted as the ratio of their respective activation fluxes over time. Line colors differentiate between ligands, whereas blue (MAPK) and green (PI3K) areas indicate the respective dominant pathway activating P70S6K. (**b**) Dominating signaling fluxes without any ligand or drug treatment, as supported by the data. PI3K signaling forms the main signaling branch in MCF7, T47D and SKBR3 (highlighted in green), while MAPK signaling dominates in MDA-MB-231 cells, including a negative feedback loop on c-RAF and RAS (highlighted in blue). Asterisks indicate activating mutations in the respective components (note: *PIK3CA*, component of the the PI3K complex, is mutated in MCF7 and T47D).

## Data Availability

Time-resolved proteomics and phospho-proteomics data used for the calibration of the mechanistic model are available at https://doi.org/10.5281/zenodo.5879246 (accessed on 9 May 2022)). The phenotypic data from the viability screen are provided as Appendix A. The model was implemented in the open-source R package dMod (https://github.com/dkaschek/dMod (accessed on 9 May 2022)), which provides an environment for the development of ODE models, parameter estimation and uncertainty analysis [94]. The ODE model we developed in the present study is available on BioModels [95] under the identifier MODEL2205030001 in SBML and PEtab formats [96].

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
