# Peer review of "Disentangling ERBB Signaling in Breast Cancer Subtypes—A Model-Based Analysis"

_cancers, 2022, doi:10.3390/cancers14102379_

Round 1
Reviewer 1 Report
The paper written by Kemmer et al. ‘Disentangling ERBB Signaling in Breast Cancer Subtypes – a Model-Based Analysis’ assesses the wiring of ERBB receptor signaling in breast cancers. The authors integrate quantitative experiments and mechanistic modeling to study the effect of ligand-drug combinations on the ERBB signaling network in different breast cancer subtypes. The model was trained on the luminal cell lines MCF7 and T47D, recapitulates time-resolved total and phosphoproteomics, and in particular, correctly predicts the proliferative capacity in the presence of different drugs and ligands based on the predicted signaling dynamics. It provides insights into the ERBB signaling activities and pointed at a shift in activated signaling branches between different breast cancer subtypes: luminal A, HER2+, and triple-negative. Summarizing, it is a good study and this work will have an impact not only on computational modeling but also on cancer biology. However, there are some things that need to be improved prior to publication.
Major comments:
- Lines 67-70 ‘However, the effect of ligand-drug combinations on the differential regulation of the ERBB, MAPK, and PI3K signaling network has not been studied in combination with individual breast cancer subtypes so far.’
For example, Shoeberl et al. explored the optimal way to therapeutically inhibit combinatorial, ligand- induced activation of the ERBB–PI3K axis (doi: 10.1126/scisignal.2000352) and recently Imoto et al. developed a modeling framework to predict potential drug targets in ERBB, MAPK, PI3K signaling network in breast cancer patients (doi: 10.1016/j.isci.2022.103944). I partially agree with this statement, however, the authors should introduce a couple of studies which investigated the drug-response mechanisms in ERBB signaling network using mathematical modeling. - Lines 249-251 ‘This signaling model represents a mechanistic description of growth-factor induced signal transduction at the receptor module towards the two branches of the downstream module, i.e., PI3K as well as MAPK signaling.’
The model developed in this study omits several key molecular interactions in the MAPK signaling triggered by the activation of ERBBs, e.g., GRB2-SOS1 recruitment to the plasma membrane, RAS-GDP to -GTP exchange by SOS1, and RAF dimerization (doi: 10.1038/35052073; PMID: 11325826). Although the current model seems to be complex enough to recapitulate experimental observations, the authors should explain the simplification of the process in the model and why it does not affect so much on the model-based predictions. - Lines 347-348 ‘Our analysis showed that the AUC of pAKT combined with the AUC of pP70S6K had the highest predictive power.’
Please add the figure or table that support this statement: quantify and compare the predictive power of signaling features, i.e., the area under the curve, the half-life, and the peak height, in ppERK, pAKT, and pP70S6K. - The details of flux analysis are not given in the method section. Please describe.
- Related to above, the relationship between the ODE model and flux analysis is unclear. The authors suggested that cell types have different feedback regulatory mechanisms, but the ODE model does not have such a regulatory loop. In ODE models, many parameters are common in different cell lines (according to the authors), so it indicates that difference in the fluxes may be originated from the initial values of the model. However, it was not clear from the analysis. Explain why they showed different fluxes and why they showed different feedback loops.
- Th methods of model reduction was not clearly understood. Please explain in details.
Minor comments:
- In Materials and Methods 2.9. Growth predictions, the authors describe the combinatorial drug effects by (1/(1+k_drug1 ))∙(1/(1+k_drug2 )). With this term, two drugs always seem to have synergistic effects on the formation of active dimers and not to work in antagonistic directions. Although the model-based proliferation predictions agreed well with experimental measurements in Figure 4c, the authors need to clarify the assumptions used in the description for the combinatorial effects of two inhibitors.
- Figure legend 3: Model calibration on time-resolved quantitative RPPA data.
Please explain RPPA again here, i.e., reverse phase protein array (RPPA). All abbreviations should be reintroduced in each figure legend as each figure legend should be treated independently. In other words, the figure legends should be understandable even without reading the main text in the manuscript. - In the Data Availability Statement, the authors wrote the ODE model developed in this study is available at https://github.com/JetiLab/ERBBsignaling, however, I currently cannot see modeling code there. Please put the code used in this study in the repository and add usage instruction in the README file.
- Many other similar ERBB signaling models have been applied to breast cancer cell lines and patient analysis. Discuss the differences and importance of this paper.

Reviewer 2 Report
General comment: This paper investigates similarities and differences in ERBB signaling associated with various breast cancer subtypes using quantitative methods, specifically, a nonlinear ODE to represent the signaling network and signal transduction part , and a Lasso regression-based model to predict proliferation rate, upon treatment with various drugs and ligands. Experiments conducted using single drug is used for model calibration and, model’s prediction for combined drug is used for validation. Paper is very well written and is recommended for publishing.
Specific comments:
- ErbB2 overexpression commonly occur in approximately 25% of all breast cancers, such over expression of ERBB is related to transformation of transient EGF-induced signaling into sustained signaling. Recently, drug repurposing opportunities in breast cancer has gained interest. Hence, the objective of the paper to investigate the applicability of available treatment options for HER2+ to other cancer subtypes (low HER-2) is justified. However, mathematical modeling of ERBB in breast cancer, its use to represent the signaling network part and effect of HER2 overexpression on cell proliferation is a well explored area [1-5], for instance, the ERBB receptor signaling network model reported in 2006 [1], used 520 ODEs and 220 kinetic parameters. The ErbB signaling in MCF-7 breast cancer cells is reported in 2007, [2] focused on ⩽30 min dynamic behavior of all four ErbB receptors (HER1, HER2, HER3, and HER4) and the key downstream intermediates ERK and Akt in response to stimulation with the ligands epidermal growth factor (EGF) and heregulin (HRG), using an ODE model with 117 species, 235 parameters, and 96 net reactions. Similarly, in [4] to investigate significance of gene expression level in determining the network dynamics, time series data of pAkt, pERK, and pc-Fos in the MCF-7, BT-474 and MDA-MB-231 cell lines were used to predict the signaling dynamics in SK-BR-3 cell line model. The results presented in the paper under review also adds significant information to the afore mentioned area. However, the authors need to clarify this in the introduction section of the paper, by adding a short comparison on what is already been explored using mathematical models. Use of L1 regularization for reducing the weight of uninformative signal features, to have a reduced set of features is also an added advantage.
- Figure 3 (a), the representative diagram that bring together the two parts-the mathematical model and subsequent proliferation model is not clear enough to read due to the color choice. Highlighting/mentioning the nonlinear ODE, single treatment (u) as input, signal features reduction using L1 regularization, marking the final two biologic specifics that needed to adjust in the unified model of ERBB signal for breast cancer subtypes (luminal, TNBC, and HER2), along with already shown prediction of proliferation as output, may make the overall picture much clearer.
- I couldn’t spot any typos except for “beast” instead of breast in the last sentence of abstract.
References
- B. Schoeberl et al., "A Data-Driven Computational Model of the ErbB Receptor Signaling Network," 2006 International Conference of the IEEE Engineering in Medicine and Biology Society, 2006, pp. 53-54, doi: 10.1109/IEMBS.2006.259754.,
- Birtwistle, Marc R., et al. "Ligand‐dependent responses of the ErbB signaling network: experimental and modeling analyses." Molecular systems biology 3.1 (2007): 144.
- Eladdadi, Amina, and David Isaacson. "A mathematical model for the effects of HER2 overexpression on cell proliferation in breast cancer." Bulletin of mathematical biology 70.6 (2008): 1707-1729.
- Imoto, Hiroaki, Suxiang Zhang, and Mariko Okada. "A computational framework for prediction and analysis of cancer signaling dynamics from RNA sequencing data—application to the ErbB receptor signaling pathway." Cancers 12.10 (2020): 2878.
- Schoeberl, Birgit, et al. "Therapeutically targeting ErbB3: A key node in ligand-induced activation of the ErbB receptor–PI3K axis." Science signaling 2.77 (2009): ra31-ra31
Reviewer 3 Report
Respected Editor
After reading carefully, I found that the paper can be accepted if the authors handle these comments carefully
- Write abstract in such a way that can provide the comprehensive information for the readers
- Use proper punctuations at the end of each equation.
- Highlight the novelty
- Improve the organization of the paper
- Write the captions of the Figs shortly and provide the details of the figures in results and discussions
- Provide some details of the findings in conclusion section
- Improve the grammar
- Add few recent references in introduction section to improve the quality of the work *Artificial neural network scheme to solve the nonlinear influenza disease model *Numerical Investigations of the Fractional-Order Mathematical Model Underlying Immune-Chemotherapeutic Treatment for Breast Cancer Using the Neural Networks*Numerical Simulations of Vaccination and Wolbachia on Dengue Transmission Dynamics in the Nonlinear Model*Numerical treatment on the new fractional-order SIDARTHE COVID-19 pandemic differential model via neural networks*A Numerical Study of the Fractional Order Dynamical Nonlinear Susceptible Infected and Quarantine Differential Model Using the Stochastic Numerical Approach
Round 2
Reviewer 1 Report
The authors have answered all my questions sincerely.
I am happy with their responses and would like to accept the paper without any further requests.